# Efficiency of Different Solvents in the Extraction of Bioactive Compounds from *Plinia cauliflora* and *Syzygium cumini* Fruits as Evaluated by Paper Spray Mass Spectrometry

**DOI:** 10.3390/molecules28052359

**Published:** 2023-03-03

**Authors:** Vinícius Tadeu da Veiga Correia, Viviane Dias Medeiros Silva, Henrique de Oliveira Prata Mendonça, Ana Luiza Coeli Cruz Ramos, Mauro Ramalho Silva, Rodinei Augusti, Ana Cardoso Clemente Filha Ferreira de Paula, Ricardo Manuel de Seixas Boavida Ferreira, Júlio Onésio Ferreira Melo, Camila Argenta Fante

**Affiliations:** 1Departamento de Alimentos, Faculdade de Farmácia, Campus Belo Horizonte, Universidade Federal de Minas Gerais, Belo Horizonte 31270-901, Brazil; 2Departamento de Ciências Exatas e Biológicas, Campus Sete Lagoas, Universidade Federal de São João del-Rei, Sete Lagoas 35702-031, Brazil; 3Departamento de Nutrição, Pontifícia Universidade Católica de Minas Gerais, Belo Horizonte 30640-070, Brazil; 4Departamento de Química, Universidade Federal de Minas Gerais, Belo Horizonte 31270-901, Brazil; 5Departamento de Ciências Agrárias, Instituto Federal de Educação, Ciência e Tecnologia de Minas Gerais, Campus Bambuí, Bambuí 38900-000, Brazil; 6Landscape, Environment, Agriculture and Food—Instituto Superior de Agronomia, Universidade de Lisboa, 1649-004 Lisboa, Portugal

**Keywords:** jabuticaba, jambolan, peel, pulp, seed, chemical profile, PS-MS

## Abstract

Jabuticaba (*Plinia cauliflora*) and jambolan (*Syzygium cumini*) fruits are rich in phenolic compounds with antioxidant properties, mostly concentrated in the peel, pulp, and seeds. Among the techniques for identifying these constituents, paper spray mass spectrometry (PS-MS) stands out as a method of ambient ionization of samples for the direct analysis of raw materials. This study aimed to determine the chemical profiles of the peel, pulp, and seeds of jabuticaba and jambolan fruits, as well as to assess the efficiency of using different solvents (water and methanol) in obtaining metabolite fingerprints of different parts of the fruits. Overall, 63 compounds were tentatively identified in the aqueous and methanolic extracts of jabuticaba and jambolan, 28 being in the positive ionization mode and 35 in the negative ionization mode. Flavonoids (40%), followed by benzoic acid derivatives (13%), fatty acids (13%), carotenoids (6%), phenylpropanoids (6%), and tannins (5%) were the groups of substances found in greater numbers, producing different fingerprints according to the parts of the fruit and the different extracting solvents used. Therefore, compounds present in jabuticaba and jambolan reinforce the nutritional and bioactive potential attributed to these fruits, due to the potentially positive effects performed by these metabolites in human health and nutrition.

## 1. Introduction

Jabuticabeira (*Plinia cauliflora*) and jambolan (*Syzygium cumini*) are plants that belong to the Myrtaceae family, whose fruits have a sweet and astringent flavor, and a purplish color due to the presence of anthocyanins [1,2]. Studies have reported the presence of several phenolic compounds in different parts of the fruit, such as flavonoids, tannins, gallic and ellagic acids, and quercetin, among others. These phytochemicals possess several bioactivities, responsible for many beneficial activities on human health and nutrition, such as antioxidant, antimicrobial, antimutagenic, and anti-inflammatory activities [3,4,5,6,7,8].

The fruits of *Plinia cauliflora* and *Syzygium cumini* have been processed and used to develop new products in different food industry sectors and in the pharmaceutical area, such as jams, desserts, wines, teas, and microcapsules [9,10]. The fruit processing chain generates a considerable number of residual co-products, mainly represented by their peel and seeds, usually discarded and underexplored [10].

Different techniques have been used to identify bioactive compounds in plant materials. High-performance liquid chromatography with an evaporative light-scattering detector and electrophoresis in polyacrylamide gel are examples of these methodologies, however, they require extensive laboratory preparation, time-consuming analyses, and high operational costs, limiting the potential of several studies [11].

Paper-spray mass spectrometry (PS-MS) has been shown to be efficient in overcoming these limitations, as it allows the rapid acquisition of fingerprints of different matrices at a considerably accessible analytical cost and without the generation of chemical residue [12,13,14,15]. Advantages of paper-spray mass spectrometry range from higher replicability to shorter data acquisition time to stronger signal stability [12]. In this method, the compounds present in the raw material are extracted by a solvent, with the drag of the extracted analytes recorded on paper and a spray ionization utilized due to the high voltage applied [16].

The factors that influence an extraction process are diverse and may be related to the conditions and time elapsed, the temperature used during the process, and the sample–solvent ratio, mainly associated with the polarity of the metabolites [17]. Therefore, the objective of this study was to determine the chemical profiles of jabuticaba and jambolan fruits by paper spray mass spectrometry (PS-MS) and to verify the influence of using different solvents (water and methanol) in obtaining fingerprints of their parts, such as the peel, pulp, and seeds.

## 2. Materials and Methods

### 2.1. Plant Material and Chemicals

Ripe jabuticaba and jambolan fruits were collected in Paraopeba—Minas Gerais, Brazil (latitude 19° 16′ 54″ S, longitude 44° 24′ 32″ W, altitude 741 m), between August and December 2020. The fruits were harvested manually in the morning from three different matrices. After harvest, they were packed in polyethylene bags, labeled, and kept in thermal boxes until transport to the Organic Chemical Laboratory of the University Federal de São João del Rei—Campus Sete Lagoas. They were then sanitized with sodium hypochlorite and fractionated into their constituent parts: peel, pulp, and seeds (the seeds were ground in a batch mill, IKA A11 basic). The samples were stored separately in a freezer at −18 °C until analysis.

Analytical grade methanol was acquired from Neon (São Paulo, Brazil), and the ultrapure water used was Milli-Q quality.

### 2.2. Physicochemical Characterization

For the physical analyses, 150 fruits (50 from each of the three matrices) of each species were evaluated. The fruit weight (g) was determined using a digital scale (Diamond, model 500). The longitudinal diameter (mm) and the transversal diameter (mm) were obtained with the aid of a digital caliper (Insize, digital caliper 0–300 mm/0–12″) with a sensitivity of 0.01 mm. To determine the soluble solids (SS) and pH of jabuticaba and jambolan pulps, a benchtop refractometer (AAKER, model: Q767BOV-OW) and pH meter (MS Tecnopon), respectively, were used [18].

### 2.3. Obtaining Extracts

The peel, pulp, and seeds of the fruits were used to prepare samples of 1.0 g each, which were added to 10 mL, in proportion 1:10 (*m*/*v*), of the different solvents analyzed: water and methanol, separately. The samples were stirred for 30 s in a vortex mixer and then kept at rest for 60 min in the dark at room temperature (±25 °C). They were centrifuged at 3500 rpm for 15 min in a bench centrifuge, and the supernatant subsequently transferred to Eppendorf tubes and kept at −18 °C in a cold chamber until further evaluations.

### 2.4. Mass Spectrometry with Paper Spray Ionization

The chemical profiles of the peel, pulp, and seeds of the fruits under study were analyzed using an LCQ mass spectrometer (Thermo Scientific, San Jose, CA, USA) equipped with a paper spray ionization source. For analysis, 2 μL of samples and 40 μL of methanol were applied on chromatographic paper and cut into a triangular shape (equilateral 1.5 cm) coupled to the equipment. The instrumental conditions of the PS-MS analysis were source voltage equal to −3.0 kV (negative mode) and +4.0 kV (positive mode), a capillary voltage of 40 V, a tube lens voltage of 120 V, transfer tube temperature of 275 °C, and mass range of 100 to 1000 for positive and negative ionization modes. A comparison was made between the mass/charge ratios (*m*/*z*) obtained in the study with those found in literature, through fragmentation by sequential mass spectrometry to identify the compounds under analysis. The collision energy used to fragment the compounds ranged from 15 to 40 V [11,18,19].

### 2.5. Statistical Analysis

The Xcalibur software version 2.1 (Thermo Fisher Scientific Inc., San Jose, CA, USA) processed the mass spectra obtained and tabulated in both ionization modes in Excel 2016 spreadsheets. The fingerprints of the samples in the positive and negative ionization modes were arranged in an X/Y matrix, with data centered on the mean, and the analysis of the principal components was performed in the MatLab software with the aid of the PLS Toolbox.

## 3. Results and Discussion

### 3.1. Physicochemical Characterization

Table 1 presents the mean values of the physicochemical characteristics analyzed in the fruits of jabuticaba and jambolan. The results showed that jabuticaba fruits presented an average pH equal to 3.12, quite similar to that of the jambolan sample (pH 3.47).

According to Costa et al. [20], the soluble solids content is an extremely important parameter for estimating fruit quality. It is typically associated with the contents of sugars, organic acids, and other microconstituents of plant samples. This characteristic is also related to in natura consumption of these raw materials by the population, and their industrialization, due to the concentrations of nectars produced by the amount of pulp.

The ratio between the diameters was calculated (LD/TD), and the results (1.004 for jabuticaba and 1.409 for jambolan) indicated a spherical shape for jabuticaba and an ellipsoidal or oval shape for jambolan (LD/TD > 1) [18]. The mean fruit weights in this analysis corroborate the values found in other studies involving these materials, such as that of Zerbielli et al. [21] when evaluating jabuticaba and Steiner et al. [22] when studying jambolan fruit samples.

### 3.2. Chemical Profile

The analyses of the spectra of aqueous and methanolic extracts of jabuticaba and jambolan fruits in ionization modes allowed the identification of 63 compounds, comprising flavonoids (40%, *n* = 25), benzoic acid derivatives (13%, *n* = 8), fatty acids (13%, *n* = 8), carotenoids (6%, *n* = 4), phenylpropanoids (6%, *n* = 4), tannins (5%, *n* = 3), terpenes (5%, *n* = 3), sugars (5%, *n* = 3), organic acids (3%, *n* = 2), amino acids (3%, *n* = 2), and esters (1%, *n* = 1), 28 being compounds tentatively identified in the positive ionization mode and 35 in the negative ionization mode.

According to the information base, the compounds found were provisionally identified through fragmentation and comparison with data already reported in the scientific literature. The attempt to identify compounds using PS-MS in the positive ionization mode (+) distinguished eight chemical classes: flavonoids (15), carotenoids (4), sugars (2), amino acids (2), fatty acids (2), one phenylpropanoid (1), one ester (1), and one benzoic acid derivative (1), verified in the subsequent constituent fractions of peel, pulp, and seeds.

Table 2 shows that all 28 compounds identified in the positive ionization mode were present in jabuticaba, while 27 substances were found in jambolan: 5-pyranopelargonidin-3-*O*-glucoside was not detected in jambolan. In the positive mode, some flavonoids maintain an isomeric relationship and evidently could not be differentiated by their exact mass; examples are delphinidin and peonidin (*m*/*z* 303), and some carotenoids, such as 9-*cis*-β-carotene, all-*trans*-β-carotene and 13-*cis*-β-carotene (*m*/*z* 537), all-*trans*-zeaxanthin, all-*trans*-lutein and *cis*-lutein (*m*/*z* 569), and *cis*-neoxanthin and *cis*-violaxanthin (*m*/*z* 601).

Ripe fruits contain high levels of carbohydrates, such as glucose and sucrose, which can be identified by PS-MS. These nutrients were also found in several other fruits, such as cagaita [24], cambuí [18], ripe banana peel [19], grumixama [25], pequi [13] and cerrado pear [35], using identical methodologies.

In both fruits under study, the presence of flavonoids stands out, mainly derived from glycosidic conjugates, a form that confers better performance to plants, such as protection against UV radiation and microbial pathogens [36,37]. This group of secondary metabolites is responsible for several beneficial health properties, such as antioxidant, antimicrobial, antimutagenic, antihypertensive, antidiabetic, and anti-inflammatory activities [3,8,12].

Among the class of flavonoids, the anthocyanins represented by the ions at *m*/*z* 287, 303, 317, 331, 433, 625, 627, 641, and 655 stand out, totaling nine tentatively identified compounds. According to Minighin et al. [38], anthocyanins are natural pigments belonging to the large class phenolic compounds that can vary from bright red to violet/blue, being water-soluble compounds, which may explain why they are all found in the aqueous extracts evaluated.

Anthocyanins are located mainly in the peel and seeds of jabuticaba and jambolan fruits. However, they were also tentatively identified in the pulp of these materials, which have a colorless hue. This is due to the migration of these constituents, possibly as a result of fruit ripening or during sample preparation [30].

The results obtained in the present study reinforce the potential for using jabuticaba peel and other dark-colored fruits, such as jambolan, in the development of natural dyes, due to the presence of anthocyanins as the main class of flavonoids tentatively identified.

Tavares et al. [30] identified nine anthocyanins, mainly derived from delphinidin, petunidin, and malvidin, in the edible parts of jambolan. The same number was obtained in another study by Tavares et al. [8] in jambolan fruit, juice, and powder. The species of the Myrtaceae family are generally considered as good sources of anthocyanins, and the higher their content, the greater the intensity of the peel color [30].

Carotenoids were also significantly represented in jabuticaba and jambolan samples in the positive ionization mode. These substances play an important role in plant metabolism and have beneficial effects at the level on human health [39]. The carotenoids all-*trans*-*β*-cryptoxanthin, 13-cis-*β*-carotene, all-*trans*-*β*-carotene, and 9-cis-*β*-carotene, tentatively identified in jabuticaba and jambolan, were also present in *Syzygium cumini* in the study reported by Faria et al. [32].

The frozen jabuticaba peel and pulp (−18 °C) were characterized by Dessimoni-Pinto et al. [40]. According to the authors, it was possible to observe that the highest concentrations of nutrients were found in the peel of this fruit, with significant levels of natural pigments, such as certain flavonoids, fibers, and carbohydrates, represented mainly by sugars.

In terms of the nutritional potential of seeds, Fidelis et al. [3] and Khan et al. [5] reinforced the activities of these constituent fractions of jabuticaba and jambolan, respectively, by observing the influence of compounds identified in these materials, with promising antimutagenic and DNA-protective properties for *S. cumini* seeds, and antioxidant, antimicrobial, anti-hyperglycemic, anti-inflammatory, and antihypertensive activities for jabuticaba seeds.

The jabuticaba contains all 35 compounds tentatively identified in the negative ionization mode, whereas for jambolan, 33 compounds were found (gallic and abscisic acids were not detected; Table 3). In the negative mode, some flavonoids also show an isomeric relationship and evidently could not be differentiated by their exact mass; examples are provided by epicatechin and catechin (*m*/*z* 289), as well as some tannins, such as castalagin and vescalagin (*m*/*z* 933).

Table 3 shows the compounds possibly identified in the negative mode (-), which can be grouped into eight chemical classes, namely flavonoids (10), benzoic acid derivatives (7), fatty acids (6), tannins (3), phenylpropanoids (3), terpenes (3), organic acids (2), and one sugar (1), with a differential distribution among the fruit fractions under study.

The flavonoid at *m*/*z* 447, cyanidin-3-*O*-glucoside, is the most abundant anthocyanin in dark-skinned fruits [50] and was detected in both jabuticaba and jambolan, as well as in the study by Lequiste et al. [51] in freeze-dried jabuticaba peel and the aqueous extract of jabuticaba peel, by Quatrin et al. [7] in powdered jabuticaba peel, by Fidelis et al. [3] in jabuticaba seeds, and by Tavares et al. [8] in jambolan (fruit, juice, and powder).

Another group of compounds tentatively identified in the present study was that of fatty acids, with six representative compounds comprising saturated (palmitic, stearic, and lignoceric acids), monounsaturated (oleic acid), and polyunsaturated (α-linolenic and linoleic acids) fatty acids. Of these, oleic and linoleic acids play an important role in human nutrition [35]. Using PS-MS, Mariano et al. [35] also identified the presence of oleic acid in cerrado pear (*Eugenia klotzschiana* Berg), another fruit of the Myrtaceae family.

The tentatively identified phenylpropanoids are also highlighted. This group of substances has demonstrated several pharmacological activities, such as anti-inflammatory, antitumor, antibacterial, and antifungal action [18]. In addition, it is reported in the literature that many chronic non-communicable diseases, such as cardiovascular diseases, type II diabetes, and various types of cancer, have their development reduced by the preventive action of phenylpropanoids [52]. When studying the chemical profile of cambuí pulp, a fruit of the Myrtaceae family, García et al. [18] observed that benzoic acid derivatives were also one of the most predominant classes.

In terms of comparing the solvents used, fatty acids, carotenoids, terpenes, and tannins were not detected in aqueous extracts. Fatty acids and terpenes have low affinity for water; carotenoids are fat-soluble molecules and are soluble in organic solvents such as petroleum ether, methanol, and acetone [53]. According to Três et al. [54], knowledge of the solubility of carotenoids in organic solvents is of fundamental importance for the understanding of recrystallization processes using solvents.

According to Bazykina et al. [55], alcohol is more effective in extracting tannins and the anthocyanins have a comparatively higher affinity for water [56]. However, some scientific studies report that it is possible to improve the solubility of anthocyanins and other phenolic compounds by using a mixture of solvents, using aqueous solutions and organic solvents (e.g., hydroethanolic mixtures) [57,58,59].

Quatrin et al. [7] also identified castalagin/vescalagin and pedunculagin in powdered jabuticaba peel. According to these authors, hydrolysable tannins play an important role in the antioxidant capacity of jabuticaba. Furthermore, as Tavares et al. [30] reported, these compounds can cause the astringent sensation of jambolan.

Preferential solubility in each solvent is a peculiar characteristic of certain phytochemical classes, such as flavonoids, fatty acids, sugars, and other phenolics, which explains, for example, the lack of a universal extraction procedure. Studies report that high polarity solvents are more effective in extracting phenolic compounds [60].

### 3.3. Principal Component Analysis (PCA)

In addition to the attempt to identify the chemical constituents of jabuticaba and jambolan, a PCA was performed among the samples. A data matrix was generated using the presence or absence of compounds found in the peel, pulp, and seeds of the fruits, when performing a fusion of the results obtained in the two ionization modes evaluated.

The models were built by selecting two main components of analysis (PC1 and PC2), which explained 61.18% (PC1 35.87% and PC2 25.31%) of the total variability of the data related to the effect of the use of different solvents and evaluation of the different parts of the fruits (Figure 1).

Through the loads PC1 and PC2, it is possible to observe that, depending on the solvent used, the jabuticaba and jambolan samples were grouped according to their constituent fractions, which leads to the realization that there was a significant difference in terms of the use of water or methanol in extracting substances.

PC1 showed the discrimination of the fruits through its positive scores, where all the methanolic fractions were found, both from jabuticaba and jambolan. The compounds responsible for this separation were hydroxybenzoic acid-*O*-hexoside, the fatty acids palmitic, α-linolenic, linoleic, oleic, octadecanoic, licanic, stearic and lignoceric, the tannins pedunculagin, castalagin/vescalagin and potentilin, the terpenes annurcoic acid and guavenoic acid, *p*-hydroxybenzoic acid, galloyl-glucose ester, and the carotenoids 9-*cis*-β-carotene/all-*trans*-β-carotene/13-*cis*-β-carotene, all-*trans*-β-cryptoxanthin, all-*trans*-zeaxanthin/all-*trans*-lutein/*cis*-lutein and *cis*-neoxanthin/*cis*-violaxanthin (positive score), and taxifolin, caffeic acid, tryptophan and apigenin (negative score), as shown in Figure 2.

In turn, PC2 shows the distinction between the constituent fractions, mainly the peel of the two fruits (positive) and the seeds (negative) of jambolan. The following compounds responsible for this differentiation are: salicylic acid, syringic acid, hydroxybenzoic acid-*O*-hexoside, kaempferol, epicatechin/catechin, taxifolin, gallocatechin, myricetin, cyanidin-3-*O*-glycoside, *p*-hydroxybenzoic, tryptophan, caffeic acid, apigenin, cyanidin, diosmetin, delphinidin/peonidin, perlagonidin-3-*O*-glucoside, peonidin-diglucoside, delphinidin-3,5-*O*-diglucoside, quercetin-3-7-diglucoside, dihydromyricetin diglucoside and *cis*-neoxanthin/*cis*-violaxanthin (positive scores), and palmitic, linoleic, oleic and galloyl-glucose ester acids (negative scores).

## 4. Conclusions

The use of mass spectrometry in the ambient ionization mode by paper spray provided comprehensive information associated with the chemical composition of jabuticaba and jambolan fruits. Compounds present in these fruits, such as flavonoids, carotenoids, tannins, and organic acids reinforce the potential bioactive effect attributed to these materials, as well as the possibility of technological and sensory use due to their physicochemical properties.

In terms of distinction between solvents, it is concluded that both water and methanol were efficient for obtaining fingerprints of different parts of the fruits (pulp, peel, and seed). Although some metabolites have not been detected in aqueous extracts, there is the possibility of using water as a solvent to extract compounds for PS-MS analysis, as it is a non-toxic solvent with good affinity for anthocyanins, the main class of flavonoids tentatively identified in this study.

## Figures and Tables

**Figure 1 molecules-28-02359-f001:**
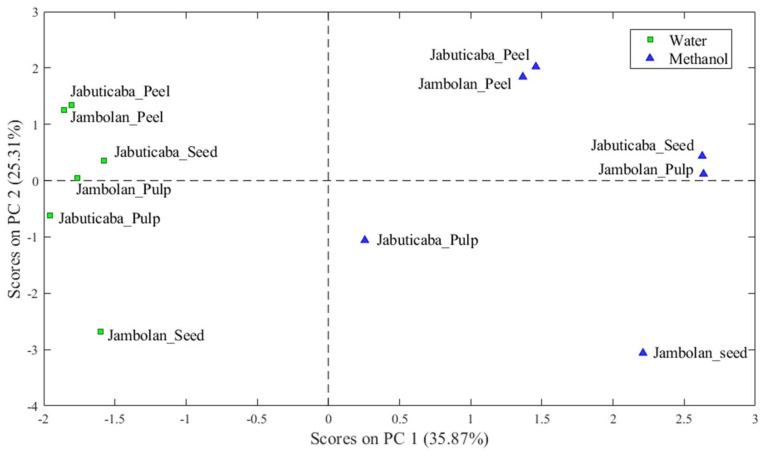
PC1 and PC2 jabuticaba and jambolan samples.

**Figure 2 molecules-28-02359-f002:**
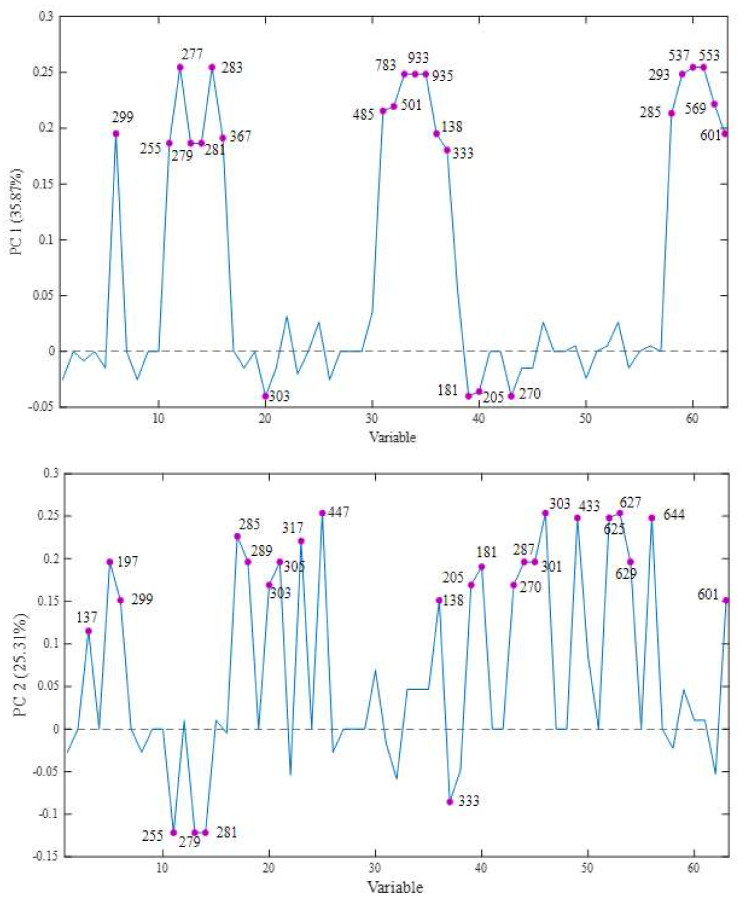
Representation of the loads responsible for the discrimination of the scores of the samples in PC 1 and PC 2.

**Table 1 molecules-28-02359-t001:** Physicochemical characteristics of jabuticaba and jambolan fruits: soluble solids (SS) (°Brix), average weight (g), transversal (mm) and longitudinal (mm) diameter.

Properties	Jabuticaba *	Jambolan *
pH	3.12 ± 0.03	3.47 ± 0.05
SS	18.93 ± 2.62	16.08 ± 2.17
Weight	5.63 ± 0.92	4.36 ± 1.04
Transversal diameter (TD)	20.51 ± 1.18	15.22 ± 1.29
Longitudinal diameter (LD)	20.59 ± 1.29	21.46 ± 0.83

* Means ± standard deviation.

**Table 2 molecules-28-02359-t002:** Compounds provisionally identified in the peel (PL), pulp (PP) and seeds (SD) of jabuticaba and jambolan fruits by (+) PS/MS.

		Jabuticaba						Jambolan							
		**Water**			**Methanol**			**Water**			**Methanol**				
*m*/*z*	**Compounds**	**PL**	**PP**	**SD**	**PL**	**PP**	**SD**	**PL**	**PP**	**SD**	**PL**	**PP**	**SD**	**MS-MS**	**Ref.**
**Benzoic acid derivatives**															
**138**	*p*-Hydroxybenzoic acid	*nd*	*nd*	*nd*	*X*	*nd*	*X*	*nd*	*nd*	*nd*	*X*	*X*	*nd*	159	[23]
**Ester**															
**333**	Galloyl-glucose ester	*nd*	*nd*	*nd*	*nd*	*nd*	*X*	*nd*	*nd*	*nd*	*nd*	*X*	*X*	**153**	[23]
**Amino acid**															
**175**	L-Arginine	*X*	*nd*	*X*	*X*	*nd*	*X*	*X*	*X*	*nd*	*nd*	*X*	*X*	70, 129	[24]
**205**	Tryptophan	*X*	*X*	*nd*	*X*	*X*	*nd*	*X*	*X*	*nd*	*X*	*X*	*nd*	188	[25]
**Phenylpropanoids**															
**181**	Caffeic acid	*X*	*X*	*X*	*X*	*X*	*X*	*X*	*nd*	*nd*	*X*	*nd*	*nd*	-	[23]
**203**	Glucose	*X*	*X*	*X*	*X*	*X*	*X*	*X*	*X*	*X*	*X*	*X*	*X*	-	[26]
**365**	Sucrose	*X*	*X*	*X*	*X*	*X*	*X*	*X*	*X*	*X*	*X*	*X*	*X*	203	[25]
**Flavonoids**															
**270**	Apigenin	*X*	*X*	*nd*	*X*	*X*	*nd*	*X*	*X*	*nd*	*X*	*X*	*nd*	-	[27]
**287**	Cyanidin	*X*	*X*	*X*	*X*	*X*	*X*	*X*	*X*	*nd*	*X*	*X*	*nd*	-	[23]
**301**	Diosmetin	*X*	*X*	*X*	*X*	*X*	*X*	*X*	*X*	*nd*	*X*	*X*	*nd*	286	[25]
**303**	Delphinidin/ Peonidin	*X*	*nd*	*X*	*X*	*nd*	*X*	*X*	*X*	*nd*	*X*	*X*	*nd*	-	[23]
**317**	Petunidin	*X*	*X*	*X*	*X*	*X*	*X*	*X*	*X*	*X*	*X*	*X*	*X*	-	[23]
**331**	Malvidin	*X*	*X*	*X*	*X*	*X*	*X*	*X*	*X*	*X*	*X*	*X*	*X*	-	[23]
**433**	Pelargonidin-3-*O*-glucoside	*X*	*nd*	*X*	*X*	*nd*	*X*	*X*	*nd*	*nd*	*X*	*nd*	*nd*	271	[28]
**501**	5-Pyranopelargonidin-3-*O*-glucoside	*X*	*X*	*X*	*X*	*X*	*X*	*nd*	*nd*	*nd*	*nd*	*nd*	*nd*	-	[25]
**611**	Rutin	*X*	*X*	*X*	*X*	*X*	*X*	*X*	*X*	*X*	*X*	*X*	*X*	303	[27]
**625**	Peonidin-3,5-diglucoside	*X*	*nd*	*X*	*X*	*nd*	*X*	*X*	*nd*	*nd*	*X*	*nd*	*nd*	301	[29]
**627**	Delphinidin-3,5-*O*-diglucoside	*X*	*nd*	*X*	*X*	*nd*	*X*	*X*	*X*	*nd*	*X*	*X*	*nd*	465	[30]
**629**	Quercetin-3-7-diglycoside	*X*	*X*	*X*	*X*	*X*	*X*	*X*	*X*	*nd*	*X*	*X*	*nd*	383	[31]
**641**	Petunidin-3,5-diglucoside	*X*	*X*	*X*	*X*	*X*	*X*	*X*	*X*	*X*	*X*	*X*	*X*	479, 317	[32]
**644**	Dihydromyricetin diglucoside	*X*	*nd*	*X*	*X*	*nd*	*X*	*X*	*nd*	*nd*	*X*	*nd*	*nd*	-	[23]
**655**	Malvidin-3,5-diglucoside	*X*	*X*	*X*	*X*	*X*	*X*	*X*	*X*	*X*	*X*	*X*	*X*	493	[33]
**Fatty acids**															
**285**	Octadecanoic acid	*nd*	*nd*	*nd*	*nd*	*nd*	*X*	*nd*	*nd*	*nd*	*X*	*X*	*X*	-	[34]
**293**	Lycanic acid	*nd*	*nd*	*nd*	*X*	*nd*	*X*	*nd*	*nd*	*nd*	*X*	*X*	*X*	257	[25]
**Carotenoids**															
**537**	9-*cis*-*β*-Carotene/ All-*trans*-*β*-carotene/ 13-*cis*-*β*-Carotene	*nd*	*nd*	*nd*	*X*	*X*	*X*	*nd*	*nd*	*nd*	*X*	*X*	*X*	-	[32]
**553**	All-*trans*-*β*-cryptoxanthin	*nd*	*nd*	*nd*	*X*	*X*	*X*	*nd*	*nd*	*nd*	*X*	*X*	*X*	-	[32]
**569**	All-*trans*-zeaxanthin/all-*trans*-lutein/ *cis*-lutein	*nd*	*nd*	*nd*	*X*	*X*	*X*	*nd*	*nd*	*nd*	*nd*	*X*	*X*	551	[25,32]
**601**	*cis*-Neoxanthin/ *cis*-Violaxanthin	*nd*	*nd*	*nd*	*X*	*nd*	*X*	*nd*	*nd*	*nd*	*X*	*X*	*nd*	-	[32]

*X*: detected, *nd*: not detected.

**Table 3 molecules-28-02359-t003:** Compounds provisionally identified in the peel (PL), pulp (PP) and seeds (SD) of jabuticaba and jambolan fruits by (-) PS/MS.

		Jabuticaba						Jambolan							
		**Water**			**Methanol**			**Water**			**Methanol**				
*m*/*z*	**Compounds**	**PL**	**PP**	**SD**	**PL**	**PP**	**SD**	**PL**	**PP**	**SD**	**PL**	**PP**	**SD**	**MS-MS**	**Ref.**
**Organic acid**															
**133**	Malic acid	*X*	*X*	*nd*	*X*	*X*	*nd*	*X*	*X*	*X*	*X*	*X*	*X*	89, 115	[24]
**191**	Citric acid	*X*	*X*	*X*	*X*	*X*	*X*	*X*	*X*	*X*	*X*	*X*	*X*	85, 111	[24]
**Benzoic acid derivatives**															
**137**	Salycilic acid	*nd*	*nd*	*nd*	*nd*	*nd*	*X*	*nd*	*nd*	*nd*	*nd*	*X*	*X*	-	[41]
**169**	Gallic acid	*X*	*nd*	*nd*	*X*	*nd*	*nd*	*nd*	*nd*	*nd*	*nd*	*nd*	*nd*	-	[42]
**197**	Syringic acid	*X*	*X*	*X*	*X*	*X*	*X*	*X*	*X*	*X*	*X*	*X*	*X*	-	[41]
**299**	Hydroxybenzoic-*O*-hexoside acid	*X*	*X*	*X*	*X*	*X*	*X*	*X*	*X*	*nd*	*X*	*X*	*nd*	-	[41]
**301**	Ellagic acid	*nd*	*nd*	*nd*	*X*	*nd*	*X*	*nd*	*nd*	*nd*	*X*	*X*	*nd*	-	[43]
**307**	Protocatechuic acid	*X*	*X*	*X*	*X*	*X*	*X*	*X*	*X*	*X*	*X*	*X*	*X*	-	[44]
**329**	Vanillic acid hexoside	*X*	*X*	*nd*	*X*	*X*	*nd*	*X*	*X*	*X*	*X*	*X*	*X*	-	[45]
**Sugar**															
**179**	Hexose	*X*	*X*	*X*	*X*	*X*	*X*	*X*	*X*	*X*	*X*	*X*	*X*	89	[24]
**Fatty acids**															
**255**	Palmitic acid	*nd*	*nd*	*nd*	*nd*	*X*	*X*	*nd*	*nd*	*nd*	*nd*	*X*	*X*	-	[26]
**277**	α-Linolenic acid	*nd*	*nd*	*nd*	*X*	*X*	*X*	*nd*	*nd*	*nd*	*X*	*X*	*X*	-	[26]
**279**	Linoleic acid	*nd*	*nd*	*nd*	*nd*	*X*	*X*	*nd*	*nd*	*nd*	*nd*	*X*	*X*	-	[26]
**281**	Oleic acid	*nd*	*nd*	*nd*	*nd*	*X*	*X*	*nd*	*nd*	*nd*	*nd*	*X*	*X*	-	[26]
**283**	Stearic acid	*nd*	*nd*	*nd*	*X*	*X*	*X*	*nd*	*nd*	*nd*	*X*	*X*	*X*	-	[26]
**367**	Lignoceric acid	*nd*	*nd*	*nd*	*X*	*X*	*nd*	*nd*	*nd*	*nd*	*X*	*X*	*X*	-	[26]
**Flavonoids**															
**285**	Kaempferol	*X*	*nd*	*nd*	*X*	*nd*	*nd*	*X*	*X*	*nd*	*X*	*X*	*nd*	197, 213, 217, 241	[25]
**289**	Epicatechin/ Catechin	*X*	*X*	*X*	*X*	*X*	*X*	*X*	*X*	*nd*	*X*	*X*	*nd*	245	[46,47]
**301**	Quercetin	*X*	*X*	*X*	*X*	*X*	*X*	*X*	*X*	*X*	*X*	*X*	*X*	-	[46]
**303**	Taxifolin	*X*	*X*	*nd*	*X*	*X*	*nd*	*X*	*X*	*nd*	*X*	*X*	*nd*	241, 285	[25]
**305**	Gallocatechin	*X*	*X*	*X*	*X*	*X*	*X*	*X*	*X*	*nd*	*X*	*X*	*nd*	137, 261	[25]
**315**	Isorhamnetin	*X*	*nd*	*X*	*X*	*nd*	*X*	*nd*	*nd*	*X*	*nd*	*nd*	*X*	300	[41]
**317**	Myricetin	*X*	*nd*	*nd*	*X*	*nd*	*nd*	*X*	*nd*	*nd*	*X*	*nd*	*nd*	-	[46]
**431**	Vitexin	*X*	*X*	*X*	*X*	*X*	*X*	*X*	*X*	*X*	*X*	*X*	*X*	-	[24]
**447**	Cyanidin-3-*O*-glucoside	*X*	*nd*	*X*	*X*	*nd*	*X*	*X*	*X*	*nd*	*X*	*X*	*nd*	285	[48]
**463**	Isoquercetin	*X*	*X*	*nd*	*X*	*X*	*nd*	*X*	*X*	*X*	*X*	*X*	*X*	-	[41]
**Phenylpropanoids**															
**311**	Caftaric acid	*X*	*X*	*X*	*X*	*X*	*X*	*X*	*X*	*X*	*X*	*X*	*X*	-	[24]
**325**	*p*-Coumaric acid hexoside	*X*	*X*	*X*	*X*	*X*	*X*	*X*	*X*	*X*	*X*	*X*	*X*	145	[25]
**339**	Cafeoil-D-glucose	*X*	*X*	*X*	*X*	*X*	*X*	*X*	*X*	*X*	*X*	*X*	*X*	-	[24]
**Terpenes**															
**263**	Abscisic acid	*nd*	*nd*	*nd*	*X*	*nd*	*nd*	*nd*	*nd*	*nd*	*nd*	*nd*	*nd*	309, 527	[49]
**485**	Annurcoic acid	*nd*	*nd*	*nd*	*X*	*nd*	*X*	*nd*	*nd*	*nd*	*nd*	*X*	*X*	441	[26]
**501**	Guavenoic acid	*nd*	*nd*	*nd*	*nd*	*X*	*X*	*nd*	*nd*	*nd*	*X*	*X*	*X*	467, 503	[26]
**Tannin**															
**783**	Pedunculagin	*nd*	*nd*	*nd*	*X*	*nd*	*X*	*nd*	*nd*	*nd*	*X*	*X*	*X*	257, 301	[46]
**933**	Castalagin/ Vescalagin	*nd*	*nd*	*nd*	*X*	*nd*	*X*	*nd*	*nd*	*nd*	*X*	*X*	*X*	301, 915	[46]
**935**	Potentillin	*nd*	*nd*	*nd*	*X*	*nd*	*X*	*nd*	*nd*	*nd*	*X*	*X*	*X*	301, 633	[46]

*X*: detected, *nd*: not detected.

## Data Availability

Not applicable.

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
