# Peer review of "Efficiency of Different Solvents in the Extraction of Bioactive Compounds from Plinia cauliflora and Syzygium cumini Fruits as Evaluated by Paper Spray Mass Spectrometry"

_molecules, 2023, doi:10.3390/molecules28052359_

Round 1
Reviewer 1 Report
The present study covers interesting information for readers and the data is presented in scientific way.
Author Response
Reviewer #1
- The present study covers interesting information for readers and the data is presented in scientific way.
R: Thank you for your comment.

Reviewer 2 Report
In this study, authors aimed to identify the chemical profiles of three fruit parts: peel, pulp, and seed of two plant species (Jabuticaba and Jambolan) by comparing the efficiency of using water and methanol.
The use of paper spray mass spectrometry can only inform the presence or absence of any compounds (in the present study 63), its detection power in terms of the resolution in quantity is well behind the high-performance liquid chromatography. However, the information within this study could be valuable for some users of the journal.
Basically figure 1 and figure 4 show the same results. I think figure 4 can be omitted from the manuscript. Figure 2 and 3 can be combined.
Author Response
Reviewer #2
- Basically figure 1 and figure 4 show the same results. I think figure 4 can be omitted from the manuscript. Figure 2 and 3 can be combined.
R: We removed Figure 4 and combined Figures 2 and 3 (Pag. 10 and lines 289 and 290). Thanks.

Reviewer 3 Report
This manuscript entitled “Efficiency of different solvents in the extraction of bioactive compounds from Plinia caulifloraand Syzygium cumini fruits as evaluated by paper spray mass spectrometry” discusses the LC-MS identification. of Jabuticaba (Plinia cauliflora) and jambolan (Syzygium cumini) fruits. The manuscript is well organized and based on solid scientific data. I strongly recommend this manuscript for publication with minor revision. My specific comments are as below,
Comments:
1. This manuscript is uneven in terms of usage of wording and font size. Please refer author guidelines.
2. Please do English grammar check for entire manuscript and sentence usage should be done properly.
3. There are many mistakes in references, many places issue no. and page no. are missing. E.g reference no: 33. 35. 37 etc.
4. I have serious concern over LC-MS identification of compounds. Since mass spec used in this study is not high resolution mass spec. Identification of the compounds based on mere molecular mass may lead to misinterpretation of the data. for many compounds MS-MS data was not given in the manuscript. I recommend author to include LC-MS data along with RT in the manuscript.
5. I recommend author to explain advantage of paper spray mass spectrometry over other techniques for analysis of natural products.
Author Response
Reviewer #3
- This manuscript is uneven in terms of usage of wording and font size. Please refer author guidelines.
R: We've made some formatting changes. We are based on the recommendations of the available document. Thanks.
- Please do English grammar check for entire manuscript and sentence usage should be done properly.
R: The grammar was checked throughout the text. Thanks.
- There are many mistakes in references, many places issue no. and page no. are missing. E.g reference no: 33. 35. 37 etc.
R: All references are cited in the text, many of them are in the tables. We check. Thanks.
- I have serious concern over LC-MS identification of compounds. Since mass spec used in this study is not high resolution mass spec. Identification of the compounds based on mere molecular mass may lead to misinterpretation of the data. for many compounds MS-MS data was not given in the manuscript. I recommend author to include LC-MS data along with RT in the manuscript.
R: Thank you for your comment. However, the objective of the work is based on the attempt to identify the compounds by PS-MS, confirmed by fragmentation and data from the literature.
- I recommend author to explain advantage of paper spray mass spectrometry over other techniques for analysis of natural products.
R: We have added this information (Lines 63-64). Thanks.

Reviewer 4 Report
I have reviewed the manuscript titled " Efficiency of different solvents in the extraction of bioactive compounds from Plinia cauliflora and Syzygium cumini fruits as evaluated by paper spray mass spectrometry" authored by Correia et.al. Although the work has been designed and executed well. I would like to suggest some amendments:
1. Introduction and conclusion is written very poorly. It must be supported by some rich review of literature.
2. Edit and proofread the entire manuscript to ensure that there are no issues related to syntax, language, and grammar.
after these corrections manuscript could be considered for possible publication.
Author Response
Reviewer #4
- Introduction and conclusion is written very poorly. It must be supported by some rich review of literature.
R: We have improved these sessions. It was added in the introduction references, a recently published article related to the characteristics of the studied fruits. Thanks.
Ref [10]: da Veiga Correia, V.T.; da Silva, P.R.; Ribeiro, C.M.S.; Ramos, A.L.C.C.; Mazzinghy, A.C.d.C.; Silva, V.D.M.; Júnior, A.H.O.; Nunes, B.V.; Vieira, A.L.S.; Ribeiro, L.V.; de Paula, A. C. C. F. F.; Melo, J. O. F.; Fante, C. A. An integrative review on the main flavonoids found in some species of the Myrtaceae family: Phytochemical characterization, health benefits and development of products. Plants 2022, 11, 2796. https://doi.org/10.3390/plants11202796.
- Edit and proofread the entire manuscript to ensure that there are no issues related to syntax, language, and grammar. After these corrections manuscript could be considered for possible publication.
R: The grammar, syntax and languague were checked throughout the text. Thanks.

Round 2
Reviewer 2 Report
The authors made significant improvements in responses to four reviewers including my suggestions. Thanks.